# A vacuum-actuated soft robot inspired by *Drosophila* larvae to study kinetics of crawling behaviour

Xiyang Sun[1], Akinao Nose[1,2], Hiroshi Kohsaka[1,3]*

**1** Department of Complexity Science and Engineering, Graduate School of Frontier Science, the University of Tokyo, Kashiwa, Chiba, Japan, **2** Department of Physics, Graduate School of Science, the University of Tokyo, Tokyo, Japan, **3** Graduate School of Informatics and Engineering, the University of Electro-Communications, Tokyo, Japan

* kohsaka@edu.k.u-tokyo.ac.jp

**Data Availability Statement:** All relevant data are within the paper and its Supporting information files.

**Funding:** This work was supported by MEXT/JSPS KAKENHI grants (17K19439, 19H04742, and

## Abstract

Peristalsis, a motion generated by the propagation of muscular contraction along the body axis, is one of the most common locomotion patterns in limbless animals. While the kinematics of peristalsis has been examined intensively, its kinetics remains unclear, partially due to the lack of suitable physical models to simulate the locomotion patterns and inner drive in soft-bodied animals. Inspired by a soft-bodied animal, *Drosophila* larvae, we propose a vacuum-actuated soft robot mimicking its crawling behaviour. The soft structure, made of hyperelastic silicone rubber, was designed to imitate the larval segmental hydrostatic structure. Referring to a numerical simulation by the finite element method, the dynamical change in the vacuum pressure in each segment was controlled accordingly, and the soft robots could exhibit peristaltic locomotion. The soft robots successfully reproduced two previous experimental phenomena on fly larvae: 1. Crawling speed in backward crawling is slower than in forward crawling. 2. Elongation of either the segmental contraction duration or intersegmental phase delay makes peristaltic crawling slow. Furthermore, our experimental results provided a novel prediction for the role of the contraction force in controlling the speed of peristaltic locomotion. These observations indicate that soft robots could serve to examine the kinetics of crawling behaviour in soft-bodied animals.

## Introduction

Over the past few decades, robotics researchers have drawn numerous inspirations from diverse animal species to design robots [1, 2]. One of the recent trends in building animal-inspired robots is to utilize soft materials to construct flexible structures, mainly because the flexibility of their body enables adaptive motions in a complex environment [2, 3]. Furthermore, the development of soft robots has shed light on the biological mechanisms of animal motion [4, 5]. In particular, soft robots are useful for understanding the kinematics of soft-bodied animals' behaviours that have a high degree of freedom and complicated dynamics [6].

20H05048 to AN and 17K07042 and 20K06908 to HK). The funders did not play any role in the study design, data collection and analysis, decision to publish, or preparation of the manuscript.

The development of biomimetic soft robots has provided valuable platforms for both robotics and neuroscience research fields by referring to animals [7], including caterpillars [1], earthworms [8], and octopi [9].

Crawling is one of the basic animal motions used to move the body in one direction. Several crawling gaits are observed in animal locomotion: two-anchor crawling (e.g. inchworms), peristalsis (e.g. fly larvae), and serpentine (e.g. snakes and nematodes) [10]. In two-anchor crawling, the animal alternatively extends and shortens its body. After the elongation, one end of the body (the head) is anchored to the ground to prevent slipping, and the other end (the tail) is released from the substrate. The subsequent shortening pulls the centre of the animal's mass forward. In peristalsis, local segmental contraction and relaxation propagate from one end to the other along the body length. The wave of segmental contraction travels in parallel (e.g. fly larvae) or antiparallel (e.g. earthworms) to the crawling direction. In serpentine crawling, waves of bending propagate along the body. These crawling gaits have been mimicked by soft robots, including worm-like robots [8], hornworm-like robots [1], snake-like robots [11], and multigait soft robots [12]. Previously, flexible braided mesh-tube structures, Meshworm and FabricWorm, have been designed based on the antagonistic muscular arrangement of earthworms [8, 13]. These robots use shape memory alloys and linear springs as actuators, respectively, and are capable of exhibiting crawling. Crawling soft robots have the potential to provide powerful tools to study the kinetics of crawling behavior. However, the use of soft robots in the analyses of animal locomotion is still limited, whereas several physical mechanisms, including how the crawling speed is realized and regulated, remain unclear [14, 15].

Larvae of fruit flies, *Drosophila melanogaster*, have provided an excellent model of a soft-bodied organism to investigate peristaltic mechanisms due to their relatively simple structure, stereotyped behaviours, and accumulated knowledge of their neural circuits [16–18]. The third instar fly larva is about 4 mm long and has a segmented body. The dominant larval behaviour is forward crawling, which propels the larval body forward by the propagation of the segmental contraction from tail to head (Fig 1; [19, 20]). Fly larvae also exhibit backward behaviour, where the segmental contraction travels from head to tail [20]. There are spike-like structures at the bottom of fly larvae called denticle bands. The majority of the denticles point

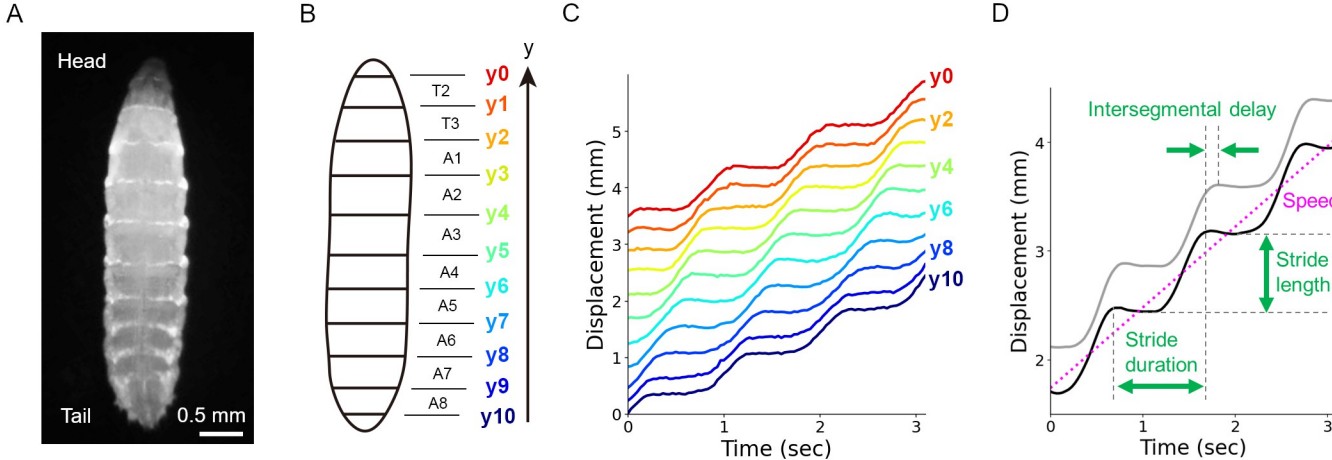

**Fig 1. Kinematic parameters in fly larval locomotion.** (A) A fluorescence image of a third-instar larva expressing a green fluorescent protein in muscle attachment sites. (B) A schematic of the segment boundaries (y0-y10) and the segment names (T2-A8). (C) Displacement of the segment boundaries during larval crawling. (D) Kinematics parameters based on segmental boundary dynamics. The black and grey curves represent the kinematics of one segment boundary and its adjacent anterior segment boundary, respectively. These panels are derived from Sun *et al.* (2022).

to the posterior, which could serve as an asymmetric friction to the substrate between forward and backward movement [21]. Recently, one soft maggot robot was designed to mimic larval muscular organization and replicate larval crawling [22]. It consisted of a series of pneumatic chambers that enabled body deformation by expansion instead of contraction, which fly larvae use. Although propagation of segmental inflation was generated in water, the ability of this previous maggot robot to perform peristaltic crawling on a solid substrate was not investigated.

In this work, we propose a new soft robot that can mimic larval crawling in reference to the properties of fly larvae. To mimic the contraction of body segments, a vacuum source and solenoid valves are used for actuation control. We implement an asymmetric interface between the robot and a ground substrate to enhance forward crawling rather than backward crawling, which replicates the denticle bands of fly larvae. This larval robot successfully exhibits a crawling motion. Furthermore, the soft robot reproduces two previous observations of the kinematics in fly larval crawling: slower speed in backward locomotion than in forward locomotion and significant involvement of segmental contraction duration in crawling speed. In addition, the soft robot provides predictions for the functional roles of kinetic parameters, including segmental contraction force and intersegmental phase delay. This study indicates that our vacuum-based soft robots could contribute to a better understanding of the mechanisms of crawling behaviour.

## Materials and methods

By mimicking the biological properties of fly larvae, we proposed a soft robot consisting of the following three components: A) a body structure with a chain of elliptical cylinder-like segments, B) a vacuum-actuated control system, and C) software for controlling and monitoring the motion of the maggot-like soft robot. Furthermore, a simulation was conducted to model and evaluate the mechanical properties of soft robots under different stresses.

### Body structure

When designing our soft larval robot, three structural properties in fly larvae were mainly considered: 1) hydrostatic structure, 2) repetitive muscular patterning, and 3) asymmetric substrate interaction via denticle bands.

Firstly, the fly larval body is filled with body fluid, and the internal pressure and the tension of the body wall play a role in supporting their body shape. There is no wall between the neighbouring segments within the larval body, so that muscle contraction in one segment decreases the volume of the segment locally, whereas each of the other segments is slightly inflated. For simplicity, we focused on the local compression during peristaltic crawling while the possible concurrent small inflation of the other segments was not implemented in soft robots. To replicate the larval hydrostatic properties, we adopted a pneumatic structure with silicone rubber among the current soft actuator candidates [2, 23]. Secondly, the configuration of muscles in the body wall is segmentally repeated in fly larvae. The larval body consists of 11 segments: three thoracic and eight abdominal segments. Although the terminal segments (the first thoracic (T1) and the last abdominal (A8) segments) have specialized structures, the other nine segments have an almost consistent structure. The larval length and width were examined as $3.69 \pm 0.56$ mm and $0.66 \pm 0.09$ mm [24]. The length-to-width ratio of a single segment in the major middle segments is about 0.5 (Fig 1). To mimic the larval shape, the individual segment was designed as an elliptical cylinder chamber with a flat plane at the bottom (S1 Fig). We constructed two soft robots with different segmental length-width ratios: The width of each segment was 30 mm, whereas the axial length of the segments of the three robots was 20 mm and

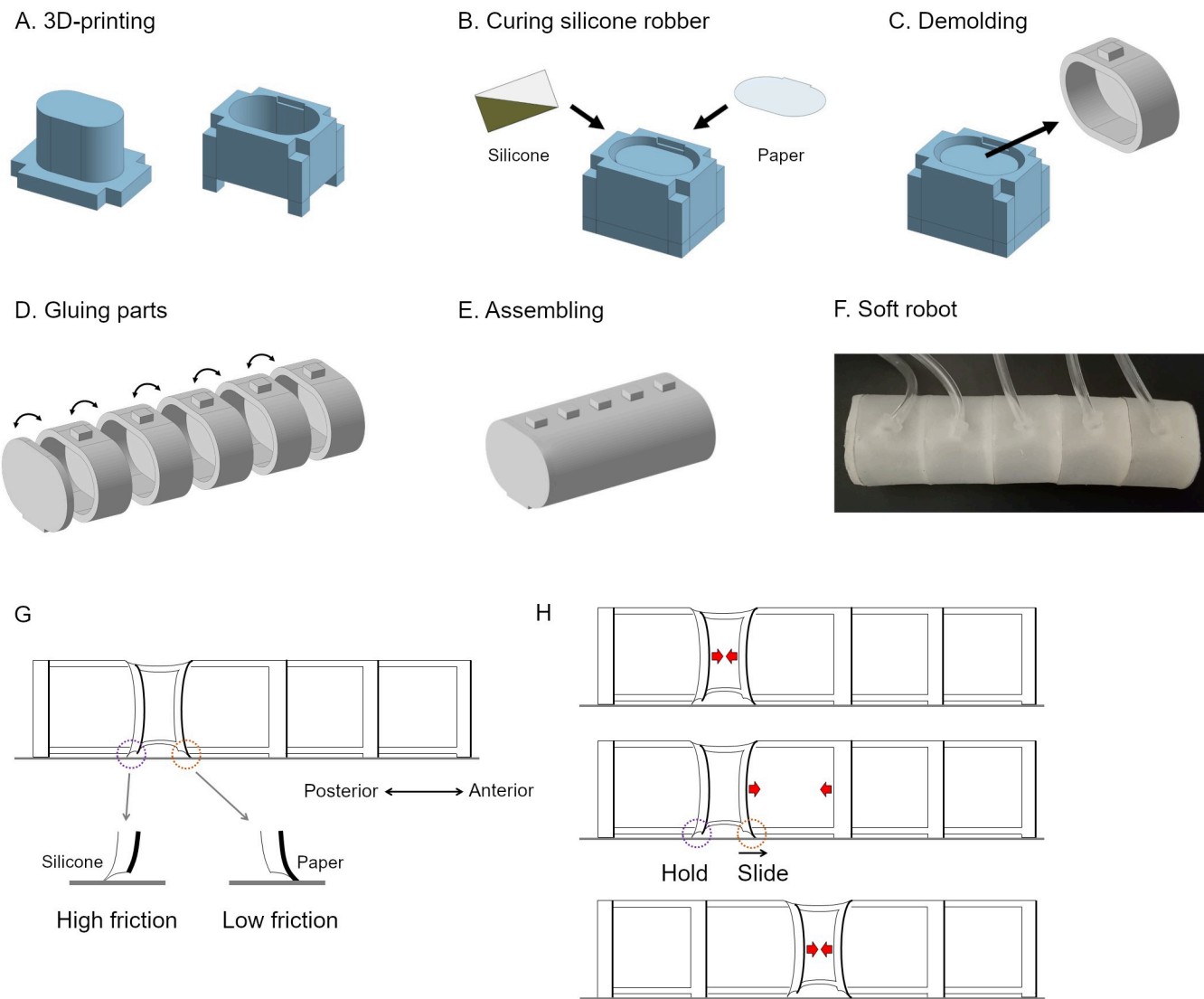

**Fig 2. The fabrication process for the soft robot.** (A) Moulds printed by a 3D printer. (B and C) Preparation for a segmental chamber of the soft robot. (D and E) Assembling segmental chambers to build the soft robot. (F) The soft robot was connected to a pneumatic pump via air tubes. (G and H) A schematic of the mechanism to generate asymmetric friction. Anterior to the right. (G) A soft robot on a substrate (grey line) exhibiting contraction of the second chamber from the posterior end. By the deformation of the chamber, the posterior edge contacts the substrate with silicone while the anterior edge contacts the substrate with paper. This configuration can make the friction at the posterior edge higher than at the anterior edge. (H) The contraction of the chambers propagates from posterior to anterior. Red arrows indicate the contraction forces by vacuum. Due to the asymmetric friction shown in G, one edge slides anteriorly while the other edge is fixed to the substrate to generate forward peristaltic crawling.

30 mm, respectively. Accordingly, their length-to-width ratios of them were 2/3 and 1, respectively. To simplify the robot while allowing us to analyze the propagation of segmental contraction, we set the number of segments in one robot as five (Fig 2). Finally, spike-like structures, named denticle bands, align at the bottom of each segment in the fly larvae. Denticle bands act as anchorage points to the ground during the propagation of segmental deformation [21]. Each denticle band normally consists of six rows of denticles in a larva. The hooked tips of four out of the six rows point posteriorly while the remaining two rows point anteriorly [21], suggesting that the friction between the ventral body surface and the ground substrate differs between forward and backward motions. To mimic this asymmetric denticle structure

between anterior and posterior directions, we implemented an asymmetric friction structure by gluing a piece of paper (Whatman paper 1001–917) at the anterior side of the segment boundary (shown as the thick lines in S1C Fig). Silicone rubber was stickier than paper; hence the physical contact between the silicone rubber and a ground substrate would generate larger friction than the one between paper and the ground substrate. By virtue of this property and the geometric fact that the angle of the segment boundary depended on the direction of the propagation of segment contraction, asymmetric friction during locomotion in different directions could be realized (Fig 2G and 2H).

We fabricated soft robots based on the design shown in Fig 2. Firstly, the moulds for the soft robots were printed with Acrylonitrile Butadiene Styrene (ABS) material with a 3D printer (WANHAO Duplicator 4S). Then, the moulds were assembled (Fig 2A), and silicone rubber (Ecoflex 00–30) was poured to fill the interior space between the inner and the outer moulds. After the gap was filled with silicone rubber, a cross-section-sized piece of paper (Whatman paper 1001–917) was placed on the top surface of the silicone rubber (Fig 2B). The rubber was cured in a drying oven (at 80˚C for about 20 minutes) and cooled down at room temperature. Afterwards, the single-segment structure was demoulded, as shown in Fig 2C. Finally, five segmental structures were glued together using the Ecoflex mixture to build the final structure (Fig 2D and 2E). A fabricated soft robot is shown in Fig 2F.

## Vacuum control system

To enable the segmental chambers to contract like body segments in fly larvae, we took advantage of a vacuum-based actuator. Fly larvae compress their segments by coordinated contraction of longitudinal and transverse muscles [20]. This motion was replicated by sucking air inside the chambers of the soft robot by vacuum. We built a system consisting of pneumatic circuits, electrical circuits (dashed and solid lines, respectively, in Fig 3), and a graphical user interface software (implemented on "Computer" in Fig 3).

The pneumatic circuits were established to control the pressure in each segment chamber. Unlike previous larvae-like robots based on segmental expansion [22], we adopted contraction as a driving force to mimic the muscular contraction of fly larvae in this study. Pneumatic pathways linked a vacuum source (TAITEC VC-15s, with a pressure range from -110kPa to 0kPa), three-way solenoid valves (ZHV 0519), vacuum pressure sensors (MPXV6115V), and robotic chambers (see "Body structure" in the Materials and methods section). Each chamber

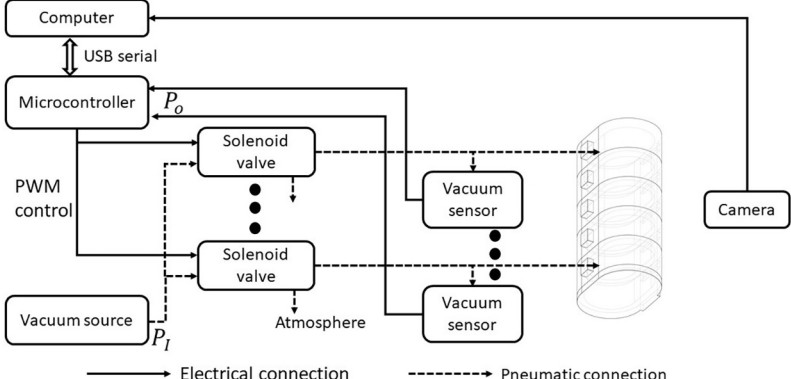

**Fig 3. The framework of the whole system.** The control system for the soft robots is composed of electrical connections (solid lines) and pneumatic connections (dashed lines).

and the vacuum source were connected through a vinyl tube and a solenoid valve. The tube was flexible and lightweight (1.4 g) compared with the soft robots (16.9–29.9 g). Even when we held the tubes, the motion of the soft robots was not disturbed, showing that the tube had little effect on the kinematics of the soft robot. The pressure within each chamber was regulated by gating these solenoid valves.

Electrical circuits were built to control the pressure valves for individual chambers. A microcontroller (Arduino Mega) was selected considering the number of Pulse-Width Modulation (PWM) pins and the output voltage. The microcontroller was connected to solenoid valves and vacuum pressure sensors to regulate and monitor robot deformation. The microcontroller's detailed electrical circuits are presented in S2 Fig, including modules for the solenoid valves and vacuum sensors. Here, an NPN transistor (TIP120) modulates an external high-power source to drive the solenoid valves based on the PWM signal from the microcontroller. The 6V external power, provided via a DC-DC power supply regulator (ARD-PWR), was the maximal operational voltage for the solenoid valves. Since the valves possessed non-negligible inductance, diodes (1N4007) were used in a flyback configuration to prevent a large back-electromotive force (back emf), which could damage these valves. These diodes could dissipate the remaining energy. The vacuum pressure, in the range of –115 to 0 kPa relative to the standard atmosphere, was measured by the output of the voltage-based vacuum sensor based on the following equation:

$$P = \frac{V_{OUT} - 0.92 \cdot V_S}{0.007652 \cdot V_S} \pm c_{Temp} \cdot P_{error} \tag{1}$$

where $P$ is the vacuum pressure, and $V_{OUT}$ and $V_S$ represent the output voltage and voltage supply, respectively, in the pressure sensor. $P_{error}$ is the pressure error, which indicates the measurement error in the pressure at a standard temperature ($P_{error}$ is in units of pressure.) Dimensional parameter $c_{Temp}$ is a temperature factor for the pressure error, which is 1 within the temperature ranges from 0 to 85°C. According to the sensor datasheet, the maximum value of $P_{error}$ was 1.725 kPa which was within the range of our experiments. Low-pass filters were applied to pressure signals from vacuum sensors (S2 Fig).

### Software for controlling and monitoring soft robot locomotion

To monitor the position of the segment boundaries and their segmental pressures in operation in real time, we designed a graphical user interface (GUI) (S3 Fig) based on Python libraries (Tkinter and pySerial). The USB camera (HOZAN camera with 12mm lens) was attached to a bracket right above the soft robot. Its frame rate was 40 frames per second, used in the following experiments. The images were sent to the computer via a USB connector, and the robotic motion could be observed on the GUI (Fig 4 right, S3 Fig left). Meanwhile, the vacuum pressures in the segmental chambers were monitored in the Sensor Signal panel of the GUI window (S3 Fig right). The position of the front end ("head") and rear end ("tail") was measured by two rulers on the platform shown in Fig 4. The soft robot crawls along the lane formed by the two rulers. Example locomotion and pressure signals are shown in S3 Fig.

### Simulation

To determine the suitable ranges of the pressure and time scale for robot deformation, we applied the finite element method (FEM) to simulate our soft robot via the commercial FEM software (Abaqus) (S1 Video). To achieve a reliable simulation, suitable viscoelastic physical models were required. In our project, the main soft body was constructed using hyperelastic material (Ecoflex 00–30). In line with previous modelling and validation work [6], the Ogden

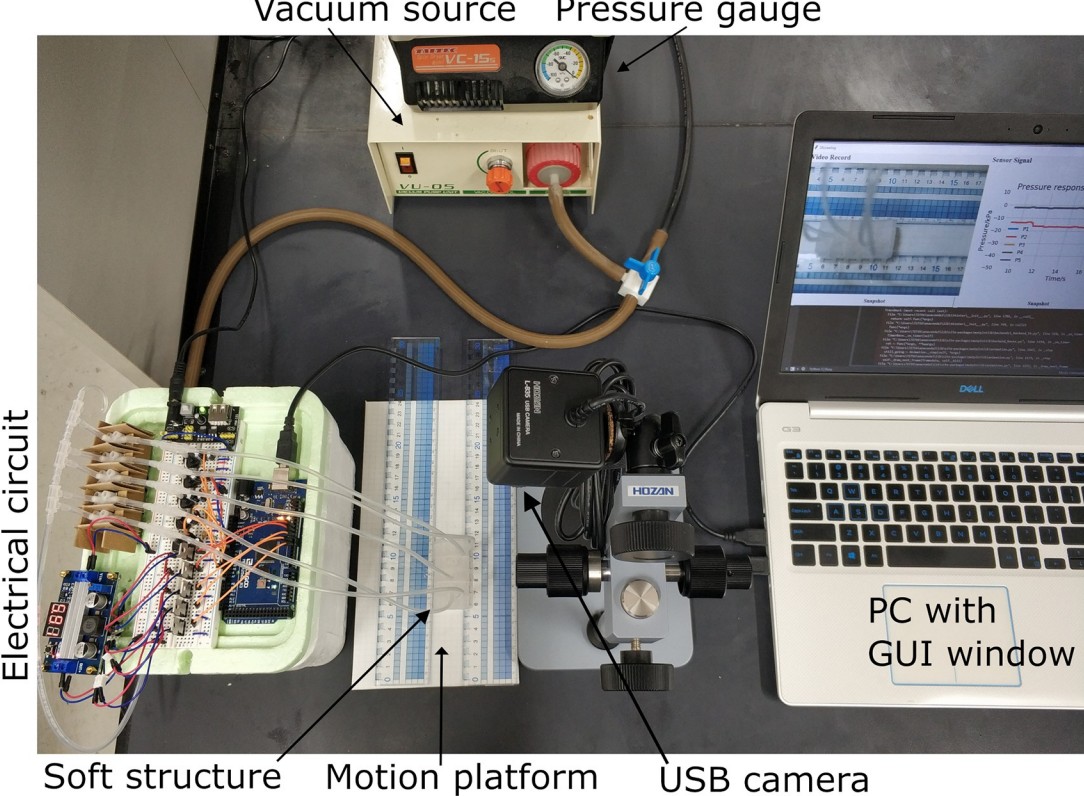

**Fig 4. System overview.** The soft robot system includes the following: the soft structure, electrical circuits, the vacuum source and pressure gauge, a motion platform, a USB camera and a PC with a GUI window.

model was adopted to model this material. The strain energy potential function $U$, which describes the elastic properties of the material, is defined as:

$$U(\lambda_1,\ \lambda_2,\ \lambda_3,\ J) = \sum_{i=1}^{N} \frac{2\mu_i}{\alpha_i^2}\left(\lambda_1^{\alpha_i} + \lambda_2^{\alpha_i} + \lambda_3^{\alpha_i} - 3\right) + \sum_{j=1}^{N} \frac{1}{D_j}(J-1)^{2j} \tag{2}$$

where $\mu_i$ and $\alpha_i$ ($i$ = 1, 2, 3) are the primary fitting parameters, and $\lambda_j$ ($j$ = 1, 2, 3) are the stretches along the x, y, and z axes. The material constant $N$ was set as three. The second summation term contains fitting parameters $Dj$ ($j$ = 1, 2, 3) to the volumetric deformation and the material Jacobian matrix $J$. We referred to the previous parameters for this hyperelastic material [6]. Meanwhile, regarding the paper (Whatman paper, see "Body structure" in the Materials and methods section) on the intersection of soft robots, the material density was 0.483 $mg/mm^3$, and its young's module was 1.71 GPa, and the Poisson ratio was -0.3 [25]. Considering the segmental deformation under vacuum pressure, we configured the self-contact condition for the interaction step. Since the physical interaction between the body and the ground surface was complicated and difficult to formalise, the frictional force was not implemented in the simulation.

We calculated the responses of the soft robot under distinct pressures using the FEM software Abaqus (Fig 5). With respect to the nonlinear dynamics within our model, we configured parameters for incrementation to ensure a successful simulation: the maximum number of increments as 100 and the minimum increment size as 1e-15. Here, the example soft robot

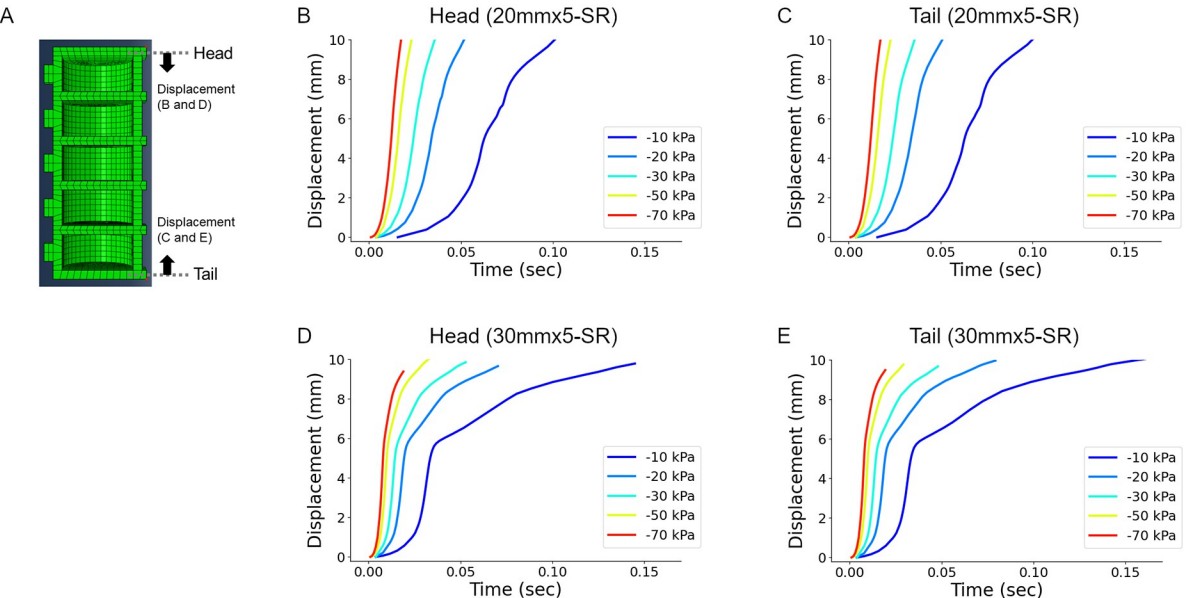

**Fig 5. Simulation of the soft robots by FEM.** (A) Section view of the soft robot generated in Abaqus. The displacements of the head and tail were monitored. (B-C) Displacements of the head (B) and tail (C) of the 20mmx5 soft robot during the simulated contraction of the corresponding segment. (D-E) Displacements of the head (D) and tail (E) of the 30mmx5 soft robot during the simulated contraction of the corresponding segment. The internal pressures were tested from -70 kPa to -10 kPa.

with five 20 mm segments was indicated as 20mmx5-SR (20 mm, five segments, and Soft Robot), and a similar shorthand notation was used for 30 mm segments 30mmx5-SR. Two points at the bottom of the terminal segment (labelled "Head" and "Tail" in Fig 5A) were used to monitor the head and tail segmental position in the longitudinal axis to record their asymmetric deformation. When the negative pressure of -10kPa was applied to the most anterior chamber (head chamber) of 20mmx5-SR, the head marker exhibited a negative displacement (Fig 5B), which indicated that the head segment was contracted, and the end of the head moved backwards. The head marker moved faster when a larger absolute value of the negative pressure (from -20kPa to -70kPa) was applied to the head chamber. This observation indicated that the kinematics of the segment dynamics could be regulated by the pressure within the soft robot chambers. Since there was no asymmetricity along the body axis in the FEM simulation, when negative pressures were applied at the most posterior chamber (tail chamber), the tail marker showed similar displacements but in the opposite direction (Fig 5C). The larger soft robot (30mmx5-SR) exhibited faster segment contraction (Fig 5D and 5E). In either case, the movement was carried out in less than 1 sec, comparable to the stride duration in fly larval locomotion.

## Statistics

Statistical calculation was conducted by Python 3.7. P-values in one-way ANOVA tests and Welch's t-test were calculated by the SciPy library.

## Results

### Control of the chambers in the soft robots by vacuum

In this study, we analyzed the locomotion of the soft robots under conditions with varied contraction forces and distinct intersegmental phases. Since the pressure of the vacuum source

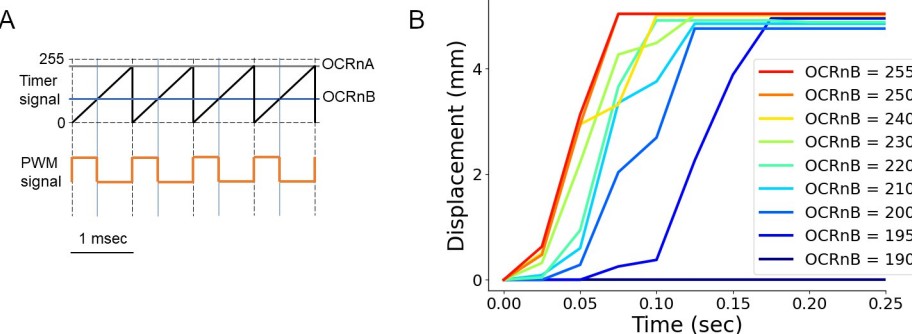

**Fig 6. Testing suitable parameter configurations in the PWM control.** (A) Schematic for the fast PWM mode. The level of OCRnA and OCRnB was set to control the frequency and duty cycle of the PWM signal, respectively. OCRnA was set to 249 to ensure a frequency of 1000 Hz. (B) Deformation of single segments with distinct duty cycles set by OCRnB.

was constant, we tried to realize varied pressures by controlling the duty cycle of the gating of solenoid valves. To this aim, we adopted the PWM method [11] to control the temporal patterns of the chamber pressures. In this method, a series of pulses are generated, and the frequency and the duty cycle of the pulses can be tuned. We set the frequency as 1000 Hz because it was fast enough to reproduce fly larval crawling that occurred on the scale of 100 ms. The timer frequency of 1000 Hz was realized by setting a parameter Output Compare Register A (OCRnA) for the Arduino Mega board as 249 (= 16 MHz / 1000 Hz / 64−1). On the other hand, the duty cycle could be adjusted by Output Compare Register B (OCRnB), another parameter in the timer. The default range of OCRnB is from 0 to 255 (Fig 6A). OCRnB set the value to threshold a sawtooth timer signal to produce 1 kHz PWM signals with varied duty cycles (Fig 6A). We then used the resulting PWM signal to control segmental deformation via vacuum pressure. The deformation of the head segment was monitored under different duty cycles (Fig 6B). The results showed that the segmental deformation could not be observed when the OCRnB is smaller than 190, which means that the solenoid valve does not work in this case. On the other hand, when OCRnB was 195 or more, the contraction of the head segment was observed. In particular, as OCRnB increased, the speed of the deformation increased, which suggested that the contraction force within the head segment chamber was higher when the duty cycle was larger. Accordingly, we succeeded in temporally controlling the pressures within the segments of the soft robot. Even when we changed the waveform from the square to others, such as sinusoidal and saw-like waveforms, the temporal profiles of chamber pressure were similar to that with a square waveform input (S4 Fig). Then we decided to use the square waveform to control the chamber pressure in the following analyses.

Referring to the results from the FEM simulation (see Simulation in the Materials and methods section), we analyzed the response of the soft robots to the pneumatic operation (Fig 7). When negative pressure was applied to the head chamber, the head marker on the 20mmx5-SR exhibited negative displacements, which was consistent with the FEM simulation results (Fig 7B). The displacement gradually reached a stable value in less than 1 second. A similar tendency was observed when negative pressures were applied to the tail chamber (Fig 7C). The larger soft robot (30mmx5-SR) showed weaker displacement than 20mmx5-SR (Fig 7D and 7E). All these deformation patterns in the measurement of the soft robots were consistent with the FEM results quantitatively. Note that the magnitude of the deformation is larger in the FEM results than in the soft robot measurements, which would be because the FEM

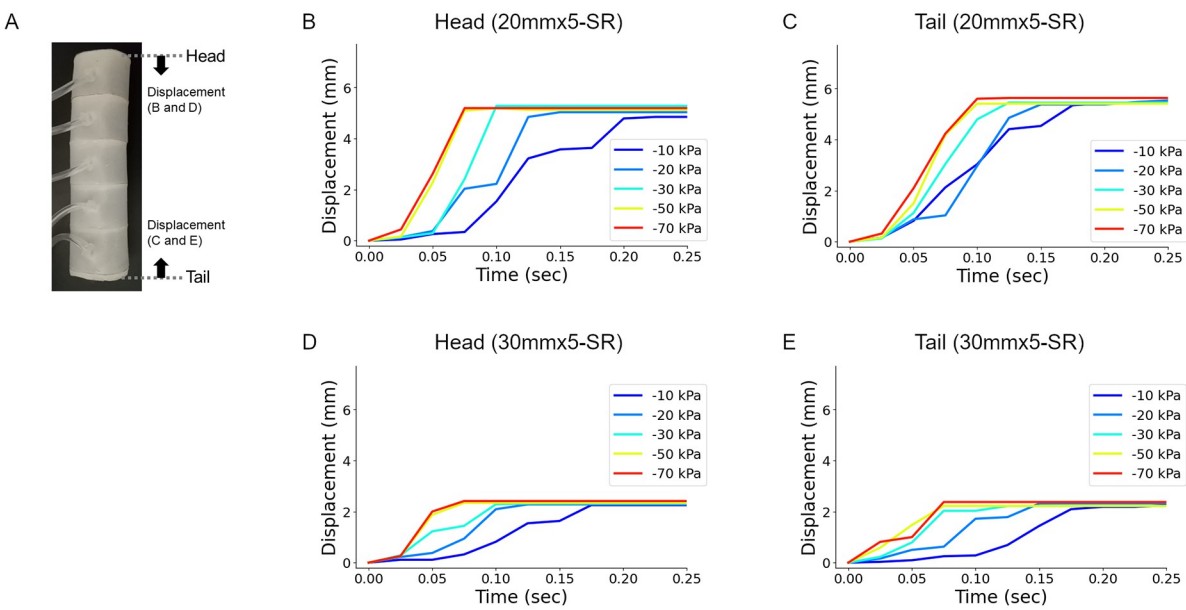

**Fig 7. Deformation of the terminal segments of the soft robots by pneumatic control.** (A) The displacements of the head and tail of the soft robots were monitored. (B-C) Displacements of the head (B) and tail (C) of the 20mmx5 soft robot by pneumatic control of the corresponding segment (head or tail segment). (D-E) Displacements of the head (D) and tail (E) of the 30mmx5 soft robot by pneumatic control of the corresponding segment (head or tail segment). The internal pressures were tested from -70 kPa to -10 kPa.

simulation lacks friction forces for model simplicity (See "Simulation" in the Materials and Methods section). These analyses indicated the larger mass of soft robots, such as 30mmx5-SR, would make the robot harder to deform and move. Hence, we described experimental results for the 20mmx5-SR soft robot in the following sections.

One striking observation is that while the displacement of the head and tail is the same in the simulation (Fig 5), the tail marker showed a slightly larger displacement upon applying negative pressure to the tail chamber than the head marker did upon the negative pressure applied to the head chamber (Fig 7). This asymmetricity in the chamber motion could be attributed to the existence of asymmetric friction in the soft robot along the body axis (Fig 2G and 2H, S1C Fig).

In summary, we implemented and optimized three properties in the soft robot to mimic fly larvae: The contraction of chambers instead of the expansion, which was used previously [22], the asymmetric feature of the interface between the soft robot and the ground substrate, and optimal ranges in pressure, time scale, and the size of the chambers.

## Robotic locomotion and its quantification

Next, the crawling ability of the soft robot was tested. Using the PWM method, we generated temporal patterns coding that negative pressure was sequentially applied from the posterior to anterior chambers (Fig 8). As an initial case, we set the vacuum pressure as –10 kPa, segmental contraction duration as one second, and the overlap of the contraction time between neighboring segments as zero (Fig 8A). Under this condition, the soft robot exhibited a crawling motion and could move forward (Fig 8B) (S2 Video). Accordingly, this result suggests that our soft robot could provide a physical model to analyze the crawling behaviour. In the following sections, we attempted to reproduce two previous experimental results and make one new prediction on fly larval locomotion with the soft robot.

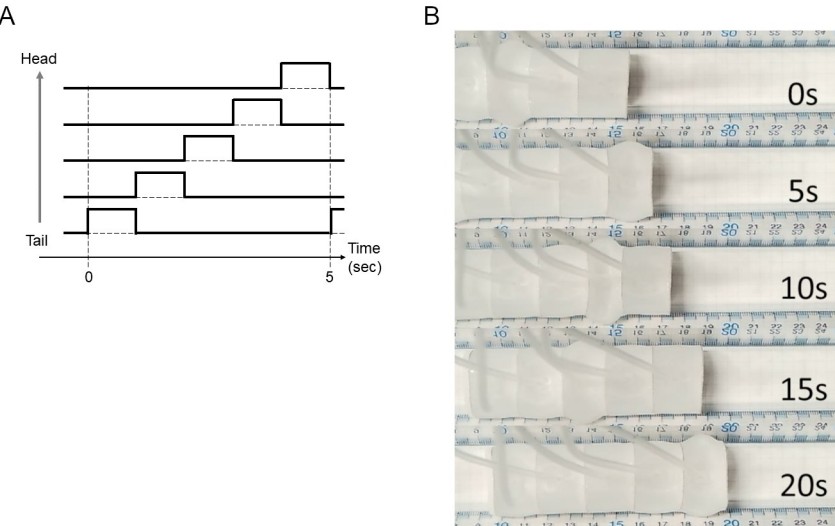

**Fig 8. Performance of the soft robots.** (A) Schematic of the control signal for the soft robot to drive forward crawling. (B) Still images of soft robot locomotion. The soft robot crawls from left to right.

## Asymmetric speed between forward and backward crawling

As the first experimental observation in fly larvae, we focused on the difference between forward and backward crawling speed. A previous study showed fly larvae move faster by forward crawling than backward [20]. However, the possible contribution of the difference in the body-ground friction during forward and backward motions to the different speeds between forward and backward crawling has not been tested. By using our soft robot, we tested this possibility. The displacements of the soft robot were calculated based on the position of its front end ("Head" in Fig 7A). The speed of robot crawling was measured using five successive strides. To generate backward crawling, we controlled the spatiotemporal pattern of the chamber pressures in a similar way to that of forward crawling (Fig 9A) but in reverse order (Fig 9B). We scanned the maximum pressures, stride durations, and intersegmental delays and compared the speed of crawling between backward and forward locomotion. In all the cases, backward crawling was slower than forward crawling (Fig 9C). This observation implies that the asymmetric properties of the interface between the larval body and the ground substrate could be a critical factor in the asymmetric crawling speed between forward and backward movements.

## Involvement of segmental contraction duration and intersegmental phase delay in locomotion speed

We next analyzed the relationship between the segmental kinematics and crawling speed. A previous study reported that a class of inhibitory interneurons (PMSIs, period-positive median segmental interneurons) was involved in crawling speed in fly larvae [26]. When the activity of PMSIs was blocked, the segmental contraction duration and the delay in contraction between neighboring segments were elongated. Furthermore, the larvae with reduced PMSI activity exhibited slower crawling. These observations suggested that the segmental contraction duration or the intersegmental phase delay should be involved in crawling speed, but this hypothesis has not been tested. Taking advantage of our soft robot, we investigated this possibility.

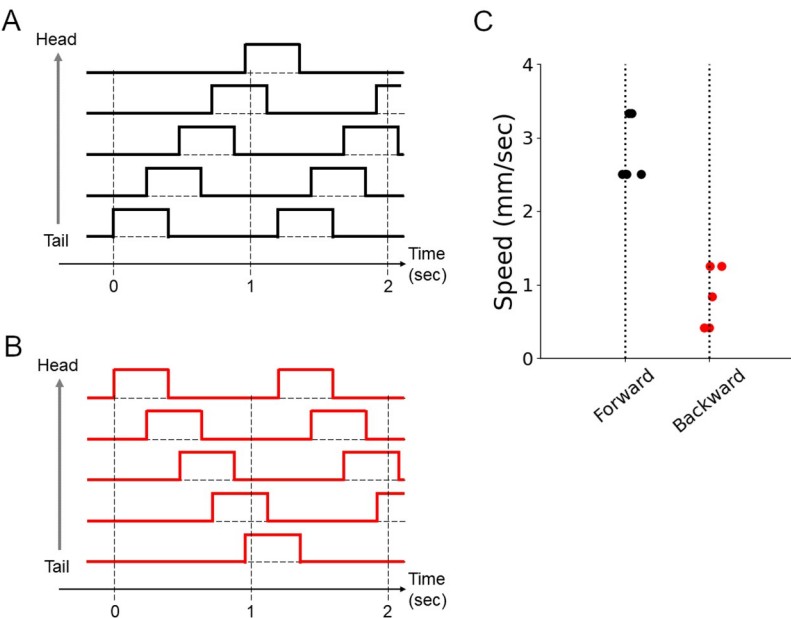

**Fig 9. Forward and backward crawling in the soft robot.** (A and B) Control signals for forward (A) and backward (B) locomotion of the soft robot. (C) Locomotion speed of the soft robot in forward and backward locomotion at the vacuum pressure of -30 kPa. Forward crawling was faster than backward crawling (p-value = 9.3 x $10^{-5}$ by Welch's t-test).

First, we changed the segmental contraction duration while keeping the minimum pressure constant (-30 kPa) and fixing the intersegmental phase delay (60%) (Fig 10A and 10B; We defined intersegmental phase delay as the ratio of an intersegmental time delay to segmental contraction duration. See below for details.) The result showed that the soft robot with a shorter segmental contraction duration (up to 0.2 sec) exhibited faster crawling (Fig 10C). This observation is consistent with the observation in the loss of function experiment of PMSIs.

Next, we perturbed the intersegmental phase delay while keeping other conditions (Fig 10D and 10E). In fly larvae, the contraction of neighboring segments overlaps during larval crawling [17]. The stride duration we measured from fly larvae was about one second, and the average segmental contraction duration was around 0.5 sec. The intersegmental delay was obtained by dividing the stride duration (1 sec) by the number of segments (10), giving a 0.1-second intersegmental delay. Accordingly, the phase delay of neighboring segments in fly larvae was 20% (= 0.1 sec / 0.5 sec). We investigated the effects of segmental phase delay on crawling speed, which has not been examined before.

We measured robot locomotion with various segmental phase delays. For our five-segment robot, 20% and 40% phase delays were too short to generate stable locomotion because all the segments were shrunk under these conditions. To make time for segments to be relaxed, we tested segmental phase delays of 60%, 80%, and 100% while keeping segmental contraction time constant (0.4 sec) and fixing the minimum pressure (-30 kPa). We found that as the segmental phase delay increased, the crawling speed became slower (Fig 10F). These phenomena would be because the smaller segmental phase delay promoted cooperative contraction of neighboring segments, leading to larger contraction of segments and faster crawling speed. To sum, the perturbation experiments with our soft robot showed that both the segmental

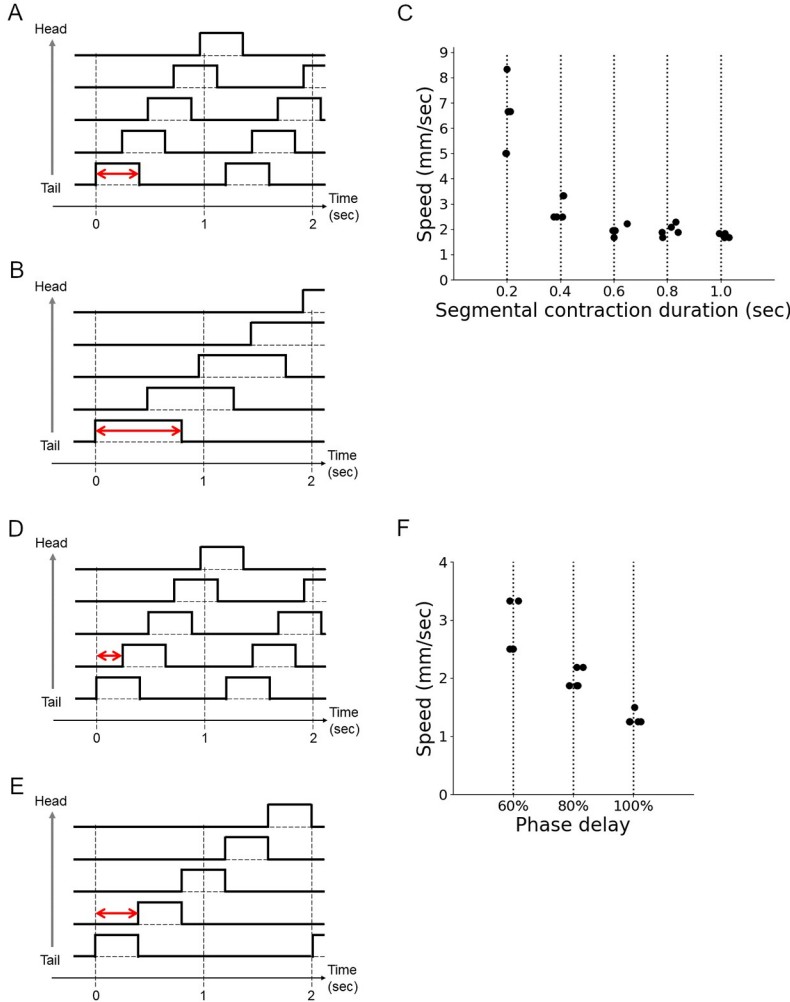

**Fig 10. Involvement of segmental contraction duration and intersegmental phase delay in crawling speed.** (A and B) Control signals with different segmental contraction duration (double-headed arrows) for the soft robot. (A) The segmental contraction duration is 0.4 sec. (B) The segmental contraction duration is 0.8 sec. The intersegmental phase delay is the same in (A) and (B). (C) Locomotion speed of the soft robot with different segmental contraction duration (p-value = $1.1 \times 10^{-4}$ by one-way ANOVA). (D and E) Control signals with different intersegmental phase delays (double-headed arrows) for the soft robot. (D) The intersegmental phase delay is 60%. (E) The intersegmental phase delay is 100%. The segmental contraction duration is the same in (D) and (E). (F) Locomotion speed of the soft robot with a different intersegmental phase delay (p-value = 0.015 by one-way ANOVA).

contraction duration and the intersegmental phase delay could be involved in crawling speed, consistent with the previous experimental observation.

## Prediction of the relationship between the maximum contraction force and locomotion speed

Finally, we analyzed the relationship between segmental contraction force and the crawling speed, which has not yet been examined in fly larvae. We systematically changed the vacuum pressure from –10 kPa to –70 kPa while keeping the segmental contraction duration constant (one second corresponding to the segmental contraction time of 0.2 sec for a five-segment soft robot) and no overlapping between neighboring segmental contractions. The results showed

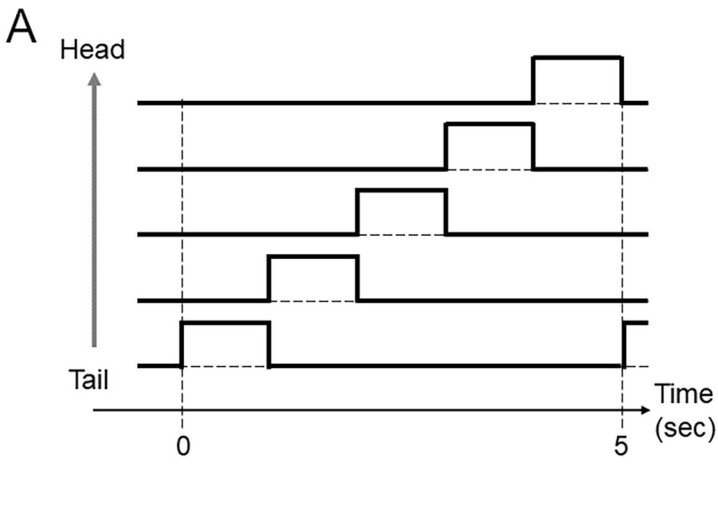

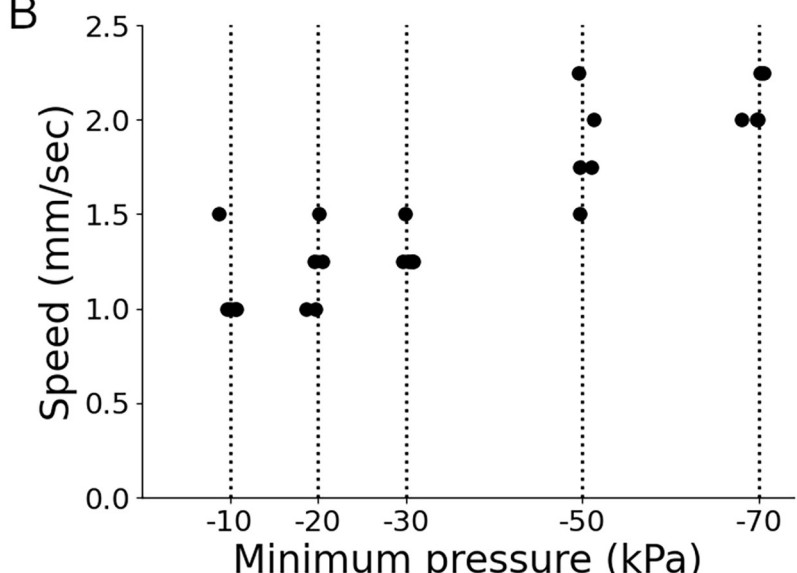

**Fig 11. Relationship between the contraction force and crawling speed.** (A) Control signals for forward locomotion of the soft robot. (B) Locomotion speed of the soft robot with different vacuum pressure amplitudes (p-value = 2.6 x $10^{-7}$ by one-way ANOVA).

that the soft robot operated with larger absolute values of the negative pressure and exhibited a faster crawling speed (Fig 11). This tendency could be observed in different segmental contraction durations. This observation provides a prediction that a larger muscular force could generate faster crawling in fly larvae.

## Discussion

In this work, we developed a new vacuum-actuated soft robot to mimic fly larval locomotion. Our soft robot could show peristaltic locomotion patterns produced by the propagation of segmental contraction waves. Two points were crucial to realizing effective peristaltic locomotion in our soft robots: First, pieces of paper inserted in the cross-section help generate asymmetric

friction between the soft robot and ground substrate, resulting in different locomotion speeds in different directions. Second, to determine the control signals for the soft robot, we used the kinematic properties in fly larval crawling and configured various kinetic parameters, including pressures, stride durations, and intersegmental phase delays. Based on the robotic locomotion, we found that the stride duration, intersegmental phase delay, and contraction force all contributed to the regulation of crawling speed.

Analyses using our soft robots were consistent and gave novel interpretations to previous works. The mechanism of PMSI neurons indicated that shorter intersegmental phase delays promoted faster speed. Assays using soft robots provided evidence (Fig 10) consistent with this. Furthermore, the observation that our soft robot crawled faster in the forward than backward direction was consistent with the previous kinematic study [20]. According to the present study, the difference in speed between forward and backward crawling would be partially attributed to the asymmetric friction property between forward and backward directions. Since two separate neural circuits are involved in forward and backward crawling in the central nervous system [27], the different forward and backward crawling speeds might be realized by two parallel mechanisms: friction asymmetricity and direction-specific neural circuits.

There are several ways to improve the performance of robotic crawling in future studies. First, the physical properties of soft materials are critical for the dynamics of soft robots. The crawling speed should be improved by utilizing soft materials with larger frictional coefficients for the soft robot. Second, a better actuation source, which can make the soft robot untethered, would broaden its applications in practical scenarios. Third, implementing soft sensors on the robotic body can make it more bionic. It has been reported that the proprioception of the body wall is key to generating an innate crawling speed in *Drosophila* larvae [28]. By monitoring the deformation on the surface of the soft robot and using this information to control the crawling behaviour, the soft robot could exhibit flexible and adaptive locomotion in various complicated environments. Last but not least, although we tested two different sizes of robots, the scale of the robot can still be modified. By changing the number and size of the segments, locomotion ability could be changed. By optimizing these conditions, the application range of our soft larval robots would be broadened.

The kinematic results derived from soft robot systems have the potential to inspire further study of larval motor outputs, including the mechanisms and kinematic effects of segmental contraction force and phase delay. Conversely, attempts to better understand the mechanisms in soft-bodied animals will contribute to designing adaptive and robust soft robots.

## Supporting information

**S1 Fig. Design of the soft robot.** (A) Mould sketch for the soft structure. (B) The segmentally-repeated structure of the soft robot. (C) Locomotion scheme with segmental deformation. Red circles show the asymmetric friction between anterior and posterior segmental boundaries in body-substrate interaction. The left is anterior. Sheets of paper inserted in the soft robot are represented by thick lines.
(TIF)

**S2 Fig. Illustration of the electrical circuits.** The left part represents the main connections for solenoid valves and pressure sensors. The right parts show the solenoid valve module (top) and pressure sensor module (bottom).
(TIF)

**S3 Fig. The GUI window for recording the locomotion of the soft robot.**
(TIF)

**S4 Fig. Single-segment deformations in 20mmx5SR by different control signals.** (A—E) The upper panels represent different control signals: square (A), sinusoid (B), and saw (C—E) waveforms. The middle panels show the pressure measured in the segmental chambers. The bottom panels represent the segment deformation induced by the control signal shown in the upper panels.
(TIF)

**S1 Video. Simulation of the soft robot deformation by Abaqus.** One chamber of the 20mmx5 soft robot is deformed by the vacuum pressure of -70 kPa.
(MP4)

**S2 Video. A video recording of soft robot locomotion.** The 20mmx5 soft robot crawls forward at the stride duration of 1 sec and by the vacuum pressure of -70 kPa.
(MP4)

## Acknowledgments

We thank Drs Jane Loveless and Dai Owaki for their critical comments on our manuscript.

## Author Contributions

**Conceptualization:** Xiyang Sun, Hiroshi Kohsaka.

**Data curation:** Xiyang Sun, Hiroshi Kohsaka.

**Formal analysis:** Xiyang Sun, Hiroshi Kohsaka.

**Funding acquisition:** Akinao Nose, Hiroshi Kohsaka.

**Investigation:** Xiyang Sun, Hiroshi Kohsaka.

**Methodology:** Xiyang Sun, Hiroshi Kohsaka.

**Project administration:** Hiroshi Kohsaka.

**Resources:** Akinao Nose, Hiroshi Kohsaka.

**Software:** Xiyang Sun, Hiroshi Kohsaka.

**Supervision:** Hiroshi Kohsaka.

**Validation:** Hiroshi Kohsaka.

**Visualization:** Xiyang Sun, Hiroshi Kohsaka.

**Writing – original draft:** Xiyang Sun, Hiroshi Kohsaka.

**Writing – review & editing:** Xiyang Sun, Hiroshi Kohsaka.

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
