## [Decision Letter · Decision Letter 0]

7 Feb 2023

PONE-D-22-28397A vacuum-actuated soft robot inspired by Drosophila larvae to study kinetics of crawling behaviourPLOS ONE

Dear Dr. Kohsaka,

Thank you for submitting your manuscript to PLOS ONE. After careful consideration, we feel that it has merit but does not fully meet PLOS ONE’s publication criteria as it currently stands. Therefore, we invite you to submit a revised version of the manuscript that addresses the points raised during the review process.

I have read through the reviewers comments and I concur with their recommendations for additions and revisions. They have also asked for some clarifications. I suggest you address the clarifications separately in an expanded response document which captures all your revisions and your specific responses to each of the reviewers' comments. 

We look forward to receiving your revised manuscript.

Kind regards,

Jit Muthuswamy

Academic Editor

PLOS ONE

Journal Requirements:

Additional Editor Comments (if provided):

Dear Dr. Kohsaka,

First of all, my apologies for the delay in getting the reviews for your manuscript and thanks for your patience. Both reviewers have recommended minor revisions and I concur. Please address the specific comments raised by the reviewers. One of the reviewers has an attachment with comments highlighted. Please make sure you address them as well.

Thanks for your patience.

Best regards

Jit Muthuswamy

Journal Requirements: 

"This work was supported by MEXT/JSPS KAKENHI grants (17K19439, 19H04742, and 20H05048 to AN and 17K07042 and 20K06908 to HK). We thank Drs Jane Loveless and Dai Owaki for their critical comments on our manuscript"

"This work was supported by MEXT/JSPS KAKENHI grants (17K19439, 19H04742, and 20H05048 to AN and 17K07042 and 20K06908 to HK). The funders did not play any role in the study design, data collection and analysis, decision to publish, or preparation of the manuscript."

Reviewers' comments:

Reviewer's Responses to Questions

**Comments to the Author**

1. Is the manuscript technically sound, and do the data support the conclusions?

Reviewer #1: Partly

Reviewer #2: Yes

2. Has the statistical analysis been performed appropriately and rigorously? 

Reviewer #1: Yes

Reviewer #2: Yes

3. Have the authors made all data underlying the findings in their manuscript fully available?

Reviewer #1: Yes

Reviewer #2: Yes

4. Is the manuscript presented in an intelligible fashion and written in standard English?

Reviewer #1: Yes

Reviewer #2: Yes

5. Review Comments to the Author

Reviewer #1: This paper takes a very interesting, bimimetic approach to better understand the propagation of motion by pneumatically controlling segmented, soft, robotic sections to emulate crawling by fly larvae. The robotic emulation of fly larvae

movement offers novel insight into how biological, neural circuits may control movement and how to design soft robots using friction asymmetry and direction specific mechanisms.

1. In the introduction, can the authors explain the differences in the

mechanics and mechanisms of peristaltic motion vs. inchworm like motion which is often used for soft robotic movement?

2. The concept of asymmetric friction using paper between segments in motion mechanics is an interesting idea to emulate the dendicle band in larvae.While the supplementary Figure 1C shows the contact areas during contractions, a free-body

diagram of forces in figure 2 might help to better understand the applied forcesand the direction of motion propagation. How does anisotropic friction arise? Does the angle of the friction force matter?

3. Why does the simulation in Fig.5 for especially for large robots not match the experiment (Fig7D&E) where displacements are smaller?

Reviewer #2: Overall, the manuscript was written well, and many different aspect were explored to obtained insight of the biological movement mechanism of fly larva. Below are some detailed comments:

Following sentence in abstract should be rewritten to be read well: and limiting the duration of segmental motor bursts slows down larval

Following sentence needs rewriting, doesn't read well: However, despite these recent advances in soft robots for studying crawling behaviour, how crawling properties, including speed, are realized and regulated remains unclear (14,15).

Authors should be more descriptive between describing the forward and backward crawling of the biological larvae, for example: sequential translation from head to tail, etc.

Expand upon following sentence: "Although coordinated motions were generated, this previous maggot robot did not realize crawling behaviour." What exactly was lacking?

Fig 1D, what is the difference between the gray and black plots, it should label or write what each are.

Mention earlier on that the biological larva features a asymmetric interface, and provide a little more detail about them. Are they active or passive, directional?

If I'm not mistaking, the volume of each chamber or the robot as a whole is not being preserved throughout the locomotion process. In other word, if you are deflating chamber(s) and releasing that air, then the the volume is not being preserved. Which is contrary to that of the biological larva, as you mentioned by the body fluid. Therefore, this difference should be mentioned in the text. Furthermore, it would be good to mention how the biological larva actuates each segments via transverse and longitudinal muscle, and how the robot actuates each segment. The differences and overalls between biological and robot larva.

Fig 4: It appears that the two rulers channel the path of the robot, if this is the case, it should be mentioned in the text.

OCRnA and OCRnB should be spelled out the first time they are used in the text.

It is mentioned " One striking difference between the simulation and the soft robot experiment was the existence of asymmetric friction." Could the asymmetric features not be added to the FEM model?

Supplementary video of the locomoting robot and FEM analysis would be beneficial for visualization.

6. PLOS authors have the option to publish the peer review history of their article (what does this mean?). If published, this will include your full peer review and any attached files.

Reviewer #1: No

Reviewer #2: **Yes: **Hosain Bagheri

---

## [Author Response · Author response to Decision Letter 0]

14 Feb 2023

Responses to the reviewers’ comments

Reviewer #1: This paper takes a very interesting, bimimetic approach to better understand the propagation of motion by pneumatically controlling segmented, soft, robotic sections to emulate crawling by fly larvae. The robotic emulation of fly larvae

movement offers novel insight into how biological, neural circuits may control movement and how to design soft robots using friction asymmetry and direction specific mechanisms.

1. In the introduction, can the authors explain the differences in the

mechanics and mechanisms of peristaltic motion vs. inchworm like motion which is often used for soft robotic movement?

We thank the reviewer for the suggestion. Following the suggestion, we described three gaits of crawling (two-anchor crawling, peristalsis, and serpentine) in the Introduction of the revised manuscript (lines 50-58 in the “Manuscript” document).

2. The concept of asymmetric friction using paper between segments in motion mechanics is an interesting idea to emulate the dendicle band in larvae.While the supplementary Figure 1C shows the contact areas during contractions, a free-body

diagram of forces in figure 2 might help to better understand the applied forcesand the direction of motion propagation. How does anisotropic friction arise? Does the angle of the friction force matter?

We appreciate the reviewer’s suggestion. The deformation of the soft robot and dynamic change in the contact angle between the soft robot and the substrate underlies the asymmetric friction. We tried to create free-body diagrams, but since the friction changes dynamically during the motion we could not create satisfactory illustrations. Instead, we made schematics of the mechanism on the asymmetric friction and demonstrated them in Figures 2G and 2H. 

3. Why does the simulation in Fig.5 for especially for large robots not match the experiment (Fig7D&E) where displacements are smaller?

We appreciate this insightful comment. As the reviewer suggested, although the simulation results are qualitatively consistent with the soft robot experiments, there are quantitative differences between the two. The main difference is that the friction force is not implemented in the simulation because the modelling frictional forces is challenging. Thus, the simulation exhibited larger deformation (Figure 5B-5E) than the soft robot experiments (Figure 7B-7E). We described this point in lines 257-259 and 334-337 in the revised “Manuscript” document. 

Reviewer #2: Overall, the manuscript was written well, and many different aspect were explored to obtained insight of the biological movement mechanism of fly larva. Below are some detailed comments:

Following sentence in abstract should be rewritten to be read well: and limiting the duration of segmental motor bursts slows down larval

We thank the reviewer for this suggestion. Following the suggestion, we revised this part as “1. Crawling speed in backward crawling is slower than in forward crawling. 2. Elongation of either the segmental contraction duration or intersegmental phase delay makes peristaltic crawling slow.” We described this in lines 30-32 in the revised “Manuscript” document. 

Following sentence needs rewriting, doesn't read well: However, despite these recent advances in soft robots for studying crawling behaviour, how crawling properties, including speed, are realized and regulated remains unclear (14,15).

We thank the reviewer for pointing out this issue. As suggested, we revised this part and described it in lines 64-67 in the revised “Manuscript” document.

Authors should be more descriptive between describing the forward and backward crawling of the biological larvae, for example: sequential translation from head to tail, etc.

We thank the reviewer for raising this point. Following the suggestion, we described the kinematics of the forward and backward larval crawling in lines 71-74 in the revised “Manuscript” document.

Expand upon following sentence: "Although coordinated motions were generated, this previous maggot robot did not realize crawling behaviour." What exactly was lacking?

We thank the reviewer for pointing out this issue. The previous maggot robot could exhibit propagation patterns of the segmental contraction in water, but the report didn’t present data about peristalsis on a solid substrate. We described this point in lines 78-82 in the revised “Manuscript” document.

Fig 1D, what is the difference between the gray and black plots, it should label or write what each are.

We thank the reviewer for raising this point. The black and grey curves show the kinematics of one segment boundary and its adjacent anterior segment boundary. We are sorry for this unclearness. We added this explanation in the legend of Figure 1D (lines 88-90 in the revised “Manuscript” document.

Mention earlier on that the biological larva features a asymmetric interface, and provide a little more detail about them. Are they active or passive, directional?

We thank the reviewer for this suggestion. They are passive and directional structures. We described the properties of the asymmetric interface in lines 74-77 in the revised “Manuscript” document.

If I'm not mistaking, the volume of each chamber or the robot as a whole is not being preserved throughout the locomotion process. In other word, if you are deflating chamber(s) and releasing that air, then the the volume is not being preserved. Which is contrary to that of the biological larva, as you mentioned by the body fluid. Therefore, this difference should be mentioned in the text. Furthermore, it would be good to mention how the biological larva actuates each segments via transverse and longitudinal muscle, and how the robot actuates each segment. The differences and overalls between biological and robot larva.

We appreciate this insightful suggestion. As the reviewer pointed out, the volume of each chamber can be changed during the locomotion. Following the suggestion, we described the difference in lines 119-124. We also described the difference in the actuation between fly larvae and soft robots in lines 180-182 in the revised “Manuscript” document.

Fig 4: It appears that the two rulers channel the path of the robot, if this is the case, it should be mentioned in the text.

We thank the reviewer for this suggestion. As suggested, we described this point in lines 232-234 in the revised “Manuscript” document.

OCRnA and OCRnB should be spelled out the first time they are used in the text.

We thank the reviewer for raising this point. Following the suggestion, we spelled out them in lines 303-305 in the revised “Manuscript” document.

It is mentioned " One striking difference between the simulation and the soft robot experiment was the existence of asymmetric friction." Could the asymmetric features not be added to the FEM model?

We appreciate this insightful suggestion. Due to the complexity of the dynamics of frictional forces, we could not implement the friction in our simulation. We described this point in lines 257-259 and 334-337 in the revised “Manuscript” document. 

Supplementary video of the locomoting robot and FEM analysis would be beneficial for visualization.

We thank the reviewer this suggestion. We included the videos of the FEM analysis and the locomoting robot as Supplementary Videos 1 and 2, respectively.

---

## [Decision Letter · Decision Letter 1]

8 Mar 2023

A vacuum-actuated soft robot inspired by Drosophila larvae to study kinetics of crawling behaviour

PONE-D-22-28397R1

Dear Dr. Kohsaka,

We’re pleased to inform you that your manuscript has been judged scientifically suitable for publication and will be formally accepted for publication once it meets all outstanding technical requirements.

Kind regards,

Jit Muthuswamy

Academic Editor

PLOS ONE

Additional Editor Comments (optional):

Reviewers' comments:

Reviewer's Responses to Questions

**Comments to the Author**

1. If the authors have adequately addressed your comments raised in a previous round of review and you feel that this manuscript is now acceptable for publication, you may indicate that here to bypass the “Comments to the Author” section, enter your conflict of interest statement in the “Confidential to Editor” section, and submit your "Accept" recommendation.

Reviewer #2: All comments have been addressed

2. Is the manuscript technically sound, and do the data support the conclusions?

Reviewer #2: Yes

3. Has the statistical analysis been performed appropriately and rigorously? 

Reviewer #2: Yes

4. Have the authors made all data underlying the findings in their manuscript fully available?

Reviewer #2: Yes

5. Is the manuscript presented in an intelligible fashion and written in standard English?

Reviewer #2: Yes

6. Review Comments to the Author

Reviewer #2: (No Response)

7. PLOS authors have the option to publish the peer review history of their article (what does this mean?). If published, this will include your full peer review and any attached files.

Reviewer #2: **Yes: **Hosain Bagheri

---

## [Editor Report · Acceptance letter]

15 Mar 2023

PONE-D-22-28397R1 

A vacuum-actuated soft robot inspired by *Drosophila* larvae to study kinetics of crawling behaviour 

Dear Dr. Kohsaka:

I'm pleased to inform you that your manuscript has been deemed suitable for publication in PLOS ONE. Congratulations! Your manuscript is now with our production department. 

Kind regards, 

on behalf of

Dr. Jit Muthuswamy 

Academic Editor

PLOS ONE